# How Many Times Can One Go Back to the Drawing Board before the Accurate Diagnosis and Surgical Treatment of Glucagonoma?

**DOI:** 10.3390/diagnostics12010216

**Published:** 2022-01-16

**Authors:** Carmen Sorina Martin, Ovidiu Dumitru Parfeni, Liliana Gabriela Popa, Mara Madalina Mihai, Dana Terzea, Vlad Herlea, Mirela Gherghe, Razvan Adam, Osama Alnuaimi, Valentin Calu, Adrian Miron, Silvius Negoita, Cornelia Nitipir, Simona Fica

**Affiliations:** 1Department of Endocrinology, Diabetes Mellitus, Nutrition and Metabolic Disorders, Carol Davila University of Medicine and Pharmacy, 020021 Bucharest, Romania; simonafica55@gmail.com; 2Department of Endocrinology, Diabetes Mellitus, Nutrition and Metabolic Disorders, Elias Emergency University Hospital, 011461 Bucharest, Romania; parfeniovidiu@gmail.com; 3Department of Oncological Dermatology, Carol Davila University of Medicine and Pharmacy, 020021 Bucharest, Romania; lilidiaconu@yahoo.com (L.G.P.); drmaramihai@gmail.com (M.M.M.); 4Department of Dermatology and Allergology, Elias Emergency University Hospital, 011461 Bucharest, Romania; 5OncoTeam Diagnostic, 030138 Bucharest, Romania; danaterzea@gmail.com; 6Department of Pathology, Fundeni Clinical Institute, 022328 Bucharest, Romania; herlea2002@yahoo.com; 7Department of Radiology, Carol Davila University of Medicine and Pharmacy, 020021 Bucharest, Romania; mirela_gherghe@yahoo.com; 8Department of Nuclear Medicine, Alexandru Trestioreanu Oncological Institute, 022328 Bucharest, Romania; 9Department of Firs Aid and Disaster Medicine, Titu Maiorescu University, 040051 Bucharest, Romania; adamrazvan31@gmail.com; 10Department of Orthopaedics and Traumatology, Elias Emergency University Hospital, 011461 Bucharest, Romania; 11Department of Radiology, Elias Emergency University Hospital, 011461 Bucharest, Romania; alnuaimi62003@yahoo.com; 12Department of Surgery, Carol Davila University of Medicine and Pharmacy, 020021 Bucharest, Romania; dramiron@yahoo.com; 13Department of Surgery, Elias Emergency University Hospital, 011461 Bucharest, Romania; 14Department of Anaesthesia and Intensive Care, Carol Davila University of Medicine and Pharmacy, 020021 Bucharest, Romania; silvius.negoita@umfcd.ro; 15Department of Anaesthesia and Intensive Care, Elias Emergency University Hospital, 011461 Bucharest, Romania; 16Department of Oncology, Carol Davila University of Medicine and Pharmacy, 020021 Bucharest, Romania; nitipir2003@yahoo.com; 17Department of Oncology, Elias Emergency University Hospital, 011461 Bucharest, Romania

**Keywords:** neuroendocrine tumor, glucagon, glucagonoma, somatostatin analogues, necrolytic migratory erythema

## Abstract

Glucagonomas are neuroendocrine tumors (NETs) that arise from the alpha cells of the pancreatic islets. They are typically slow-growing tumors associated with abnormal glucagon secretion, resulting in one or more non-specific clinical features, such as necrolytic migratory erythema (NME), diabetes, diarrhea, deep vein thrombosis, weight loss, and depression. Here, we report the case of a 44-year-old male with a history of diabetes mellitus, presenting with a pruritic and painful disseminated cutaneous eruption of erythematous plaques, with scales and peripheral pustules, misdiagnosed as disseminated pustular psoriasis and treated for 2 years with oral retinoid and glucocorticoids. During this period, the patient complained of weight loss of 32 kg and diarrhea and developed deep vein thrombosis. These symptoms, together with an inadequate response to therapy of the skin lesions, led to the reassessment of the initial diagnosis. Laboratory tests confirmed elevated plasma glucagon levels (>1000 pg/mL) and computed tomography (CT) scans revealed a 35/44 mm tumor in the pancreatic tail. Due to considerable disease complications and the COVID-19 pandemic, the surgical removal of the tumor was delayed for nearly 2 years. During this time, somatostatin analogue therapy efficiently controlled the glucagonoma syndrome and likely prevented tumor progression. As in other functional pancreatic NETs, the early clinical recognition of hormonal hypersecretion syndrome and the multidisciplinary approach are the keys for best patient management.

## 1. Introduction

Glucagonomas are neuroendocrine tumors (NETs) derived from multipotential stem cells of endodermal origin that arise from the alpha cells of the pancreatic islets [1]. In 1942, Becker et al. first described the typical glucagonoma skin eruption in a 45-year-old woman presenting with wight loss, widespread dermatitis, glossitis and abnormal glucose tolerance associated with a pancreatic tumor [2]. In 1966, McGavran and colleagues published the first report on a 42-year-old woman who presented with skin eruption, anemia and diabetes mellitus along with metastatic alpha cell tumor of the pancreatic islets and elevated plasma glucagon levels [3].

Glucagonomas are rare, with an annual incidence of 0.01 to 0.1 new cases per 100,000 [4]. The incidence is similar in both sexes and most patients present in the fifth to sixth decade of life. At diagnosis, the tumors are typically large (>3 cm), located in the tail or the body of the pancreas due to the high prevalence of alpha cells in this area and over 50% of them are metastatic [5]. Usually, glucagonomas are solitary tumors; however, less than 10% of glucagonomas are associated with multiple endocrine neoplasia type 1 syndrome (MEN1) [1]. Germline alterations in MUTYH, CHEK2, BRCA2, MEN-1 and VHL have been found in patients with seemingly sporadic pancreatic NETs (pNETs), suggesting that all patients with glucagonoma should be considered for testing for inherited genetic syndromes [6].

Glucagon is a single-chain polypeptide that comprises 29 amino acids. Bioactive glucagon is produced by enzymatic cleavage of a proglucagon precursor, and it is secreted from pancreatic alpha cells in response to hypoglycemia and increasing concentrations of amino acids, gastric inhibitory peptide and ghrelin. Glucagon increases glycogenolysis and gluconeogenesis via the stimulation of the cAMP pathway in liver cells, resulting in the elevation of plasma glucose. Hyperglycemia, insulin, somatostatin and glucagon-like peptide 1 (GLP-1) inhibit glucagon secretion [7]. Other actions of glucagon include the stimulation of lipolysis and the relaxation of smooth muscle in the stomach, duodenum, small bowel and colon [5].

Glucagonoma syndrome is caused by the excessive tumoral secretion of glucagon, and it is also called 4D syndrome because it consists of dermatosis, diabetes, deep vein thrombosis and depression [5]. Necrolytic migratory erythema (NME), first described by Wilkinson in 1971 [8], is present in 70–80% of patients with glucagonoma. It characteristically begins as erythematous papules or plaques involving the periorificial areas, intertriginous areas and extremities [4]. Often, the lesions are pruritic and painful, and they gradually enlarge and coalesce, gaining an annular appearance due to central clearing, induration, peripheral blistering and variable scaling. They resolve spontaneously in approximately two weeks. Patients also present with mucosal involvement resulting in angular cheilitis, stomatitis, glossitis and blepharitis [9]. Alopecia and nail changes, such as onychoschizia and distal onycholysis, are other frequent findings in NME patients. Hypoaminoacidemia, zinc and essential fatty acid deficiency, and the induction of inflammatory mediators in the epidermis combined with elevated plasma glucagon levels contribute to the epidermal dysfunction and skin lesions [10]. Diabetes mellitus is present at diagnosis in approximately 40% of patients [11]; however, glucose intolerance occurs in 68–95% of patients with glucagonoma. Weight loss is a common presenting feature occurring in 60–80% of patients. By diverting dietary amino acids into glucose and away from protein synthesis, the catabolic action of glucagon and glucagon-like peptides, such as GLP-1, may be the reason behind the weight loss [4]. Almost 50% of patients with glucagonoma develop venous thrombosis that usually manifests as a deep vein thrombosis or pulmonary embolism. The mechanism for thromboembolic disease is poorly understood but appears to be a consequence of an increase in factor X produced by the pancreatic alpha cells. Neuropsychiatric manifestations, such as depression, insomnia, dementia, psychosis, paranoid delusions, hyperreflexia, ataxia, agitation, optic atrophy and proximal muscle weakness occur in 20% of patients [4]. Elevated plasma glucagon levels and co-secretion of gastrin, VIP, serotonin or calcitonin may cause diarrhea [5].

Fasting plasma glucagon levels are usually elevated 10-to-20-fold in patients with glucagonoma (normal < 50 pg/mL). Plasma glucagon levels over 500 pg/mL are highly suggestive and above 1000 pg/mL are virtually diagnostic of glucagonoma [4]. Conditions such as sepsis, abdominal surgery, Cushing’s syndrome, acute pancreatitis, trauma, hypoglycemia, fasting, renal and hepatic failure induce moderate elevations in the serum glucagon concentration (<500 pg/mL) [11]. In the presence of the glucagonoma syndrome, a positive diagnosis is likely even in the presence of a glucagon concentration below 500 pg/mL [4].

Imaging studies are required for tumor localization and staging of the disease. Multiphasic contrast-enhanced computed tomography (CT) scans have high sensitivity for detecting intrapancreatic NETs, while magnetic resonance imaging (MRI) has a better sensitivity for liver metastases [5]. Other imaging studies with high specificity and sensitivity, such as endoscopic ultrasound (EUS), which is also used for transmucosal needle biopsy (EUS-FNA), somatostatin-receptor (SSTR) scintigraphy (SRS) and SSTR-positron emission tomography (PET) imaging with 68-gallium-DOTA-D-Phe1-Tyr3-octreotate (Gallium Ga-68 DOTATATE), Ga-68 DOTATOC or 64-Cu-DOTATATE PET/CT, respectively, should be performed in order to identify the tumor and small metastases, including those outside of the abdominal region [4,12,13,14].

Surgical resection is the only curative option when the tumor is localized [15]. Nutritional support consisting of intermittent infusions of amino and fatty acids may reverse the catabolic effects of elevated glucagon level and have been associated with long-term resolution of NME [4,16]. Somatostatin analogs (SSA), such as octreotide and lanreotide, inhibit the secretion of glucagon, improve NME, diabetes, diarrhea and neurological symptoms, and significantly lengthen the time to tumor progression [5]. The most frequent site of metastatic involvement is the liver. In patients with limited liver tumor burden, no extrahepatic metastases and normal liver function, hepatic resection is recommended [17]. Almost half of pNETs present as metastatic disease at diagnosis. Often, the liver is the only site of metastases. There are three categories of neuroendocrine liver metastases: single metastasis of any size (type I), isolated metastatic bulk accompanied by smaller deposits, with both liver lobes always involved (type II); and disseminated metastatic spread, with both liver lobes always involved (type III). For patients with type I metastatic pNETs, grade 1-2 (G1–G2) and in the absence of extra-abdominal metastatic disease, radical resection is associated with good outcomes and significantly better survival rates when compared to patients who undergo medical treatments alone [15]. Palliative techniques, such as radiofrequency ablation and cryoablation, hepatic artery embolization via the infusion of Gelfoam powder, in conjunction with chemotherapy (doxorubicin, cisplatin or streptozocin) or by infusion of radioactive isotopes (yttrium-90 (90-Y)), are used in the management of advanced metastatic disease [4]. Systemic chemotherapy and molecular targeted agents are reserved for advanced progressive tumors. Peptide receptor radioligand therapy (PRRT) with a radiolabeled SSA has demonstrated efficiency due to the unusually high expression of SSTR in glucagonomas [5].

Glucagonomas are typically advanced and metastatic by the time of diagnosis; however, the disease is slow to progress. Age, grade and distant metastases are the most significant predictors of survival [4]. The National Comprehensive Cancer Network (NCCN) guidelines recommend an initial 3-to 12-month post-resection follow-up with history and physical examination, serum glucagon level, abdominal multiphasic CT or MRI and chest CT scan, and a 6-to 12-month follow-up thereafter [18].

## 2. Case Presentation

In September 2017, a 44-year-old male was admitted to our department of dermatology for cheilitis, stomatitis and a widespread skin eruption consisting of red papules, papulo-vesicles and erythematous erosive plaques, with irregular, slightly elevated pustular and scaling borders with varying diameters ranging from 5 mm to 4 cm, some covered with crusts, that tended to coalesce into large, circular, eroded areas, preferentially distributed on the buttocks, groin, lower and upper limbs, and also on the trunk, genital and perineal area (Figure 1). Pruritus and pain accompanied the cutaneous lesions. Diffuse alopecia, a grayish hair discoloration and onychodystrophy were also noticed. A year earlier, in a territorial department of dermatology, the same skin lesions had been erroneously diagnosed as disseminated pustular psoriasis and had been unsuccessfully treated with oral retinoid, medium glucocorticoid dosages and the skin grafting of a large cutaneous necrotic lesion on the anterior face of the left leg.

The patient had an unremarkable family history but an important personal history for his age. In 2008, he had been diagnosed with diabetes mellitus, which was initially treated with oral antidiabetic medication and more recently with insulin therapy using the basal bolus regimen, as metabolic control could not be obtained partly due to the corticosteroid treatment for the alleged psoriasis. Additionally, in 2016, he developed an extensive deep vein thrombosis of the left lower extremity comprising the external iliac, the common and superficial femoral, and the popliteal veins. Furthermore, in the last 2 years, the patient reported recurrent episodes of diarrhea and an unintentional weight loss of 32 kg. At this particular turning point in his story, our dermatology department reconsidered the diagnosis as a necrolytic migratory erythema and, for the first time, integrated his skin condition with a syndromic disease, glucagonoma syndrome. The case was discussed in our multidisciplinary team (MDT), and we agreed on further diagnostic investigations as the clinical hormonal syndrome of excessive glucagon secretion was highly probable.

Unfortunately, in October 2017, the patient fell from standing height and suffered a left pertrochanteric comminuted fragility fracture, and osteosynthesis with gamma nail was performed (Figure 2a–c). The immediate evolution was further complicated by wound dehiscence that called for surgical debridement.

In February 2018, upon presentation to our department of endocrinology, the clinical examination revealed a middle-aged male patient unable to walk unassisted or without a frame, with severe bilateral hip arthralgia, slightly elevated blood pressure (150/90 mmHg) and tachycardia (heart rate = 110 bpm). The laboratory tests revealed normocytic normochromic anemia (Hb = 11 mg/dL), hypoproteinemia (5 g/dL), elevated inflammatory markers (ESR = 91 mm/h, fibrinogen = 668 mg/dL, high sensitivity CRP = 73.7 mg/L), high glycated hemoglobin (HbA1c = 7.6%), normal chromogranin A (CgA) and 5-hydroxyindoleacetic acid (5-HIAA), and elevated plasma glucagon levels (2130 ng/L, N < 209) that confirmed the positive diagnosis of glucagonoma syndrome. Biochemical screening for MEN 1 syndrome was negative (serum calcium = 8.3 mg/dL, PTH = 32.62 pg/mL, 25 hydroxy vitamin D = 27.27 ng/mL, prolactin = 12 ng/mL, IGF1 = 114 ng/mL). A multiphasic contrast-enhanced CT diagnosed a 35/44 mm tumor with heterogenous texture located in the pancreatic tail, multiple peritumoral nodules suggestive of locoregional metastases, two incompletely consolidated fractures at the sixth and seventh ribs, and a reduction in bone density and the height of the thoracolumbar vertebras (Figure 3a–c).

Tc-99m-methylene diphosphonate (MDP) bone scintigraphy indicated an abnormally high accumulation of the radiotracer throughout thoracolumbar spine and focal in the right scapula, C5 and C9 right ribs, C12 left rib, L2 vertebra and the left femoral head. The case was rediscussed in our MDT and pancreatic surgery was postponed for further investigations. The glucocorticoid treatment was tapered down and ultimately interrupted, and long-acting SSA therapy with lanreotide Autogel 120 mg every 28 days was started. In a matter of weeks, the NME lesions and diarrhea waned, and the biochemical parameters improved. The glycemic targets were easily achieved with low dosages of long-acting insulin (HbA1c = 6.1%), the inflammatory syndrome ameliorated (ESR = 30 mm/h, fibrinogen = 479 mg/dL) and plasma proteins returned to their normal range (6.4 g/dL).

A spine MRI exposed multiple vertebral fractures at T11, T12, L1, L2 and L4 levels and no osseous spinal metastases (Figure 4). As dual energy X-ray absorptiometry (DXA) also displayed a low bone mineral density for his age (right total hip Z score = −2 SD), the patient was diagnosed with severe secondary glucocorticoid-induced osteoporosis and received treatment with intravenous zoledronic acid.

For disease staging, we performed whole body 18F fluorodeoxyglucose-positron emission tomography (FDG-PET)/CT, which revealed a 34/38 mm hypermetabolic (standardized uptake value (SUV) = 6.67) tumor in the pancreatic tail and satellite solid tumors nearby the primary tumor with no uptake of F-18 FDG. There was no imaging proof of systemic metastases. Using whole body 99mTc-EDDA/HYNIC-Tyr3-Octreotide scintigraphy (SRS) with single photon emission tomography (SPECT-CT) acquisition we noticed a 41/33 mm well-circumscribed pancreatic tumor with high radiotracer uptake. The pancreatic tumor was the only situs for the radiotracer uptake (Figure 5).

For further tumor characterization, we decided to use EUS, which also confirmed a 35/39 mm hypoechoic, heterogenous mass in the tail of the pancreas that expanded into the splenic hilum with peripheral hypervascularization. The EUS-FNA and pathological analysis revealed nests of uniform round cells with monotonous nuclei, without evident mitoses in hematoxylin and eosin staining. Immunohistochemical staining was positive for CgA, synaptophysin, and cytokeratin AE1/AE3 with a Ki67 proliferation index of 2%. As far as we knew, we were facing a TNM stage III, well-differentiated, low grade, G1, sporadic, functional pNET with regional lymph node involvement [18].

The six month follow-up in September 2018 was uneventful, with the glucagonoma syndrome well controlled on SSA therapy, normal plasma CgA, decreasing glucagon levels (863 ng/L) and stable disease on CT (Figure 3d). It is important to mention that the patient received treatment with vitamin K antagonists for the deep vein left leg thrombosis and displayed high fluctuations in INR values. Despite our plan for the tumor surgical resection, the preoperative biological check-up revealed an INR value of 14, which prompted intravenous phytonadione administration. The INR reached the therapeutic range; however, the patient developed sagittal sinus thrombosis that was eventually favorably managed by the department of neurology.

In the spring of 2019, the patient maintained clinical remission of the hormonal syndrome and stable disease imaging. Nonetheless, he presented with a purulent cutaneous fistula at the incision level performed for the pertrochanteric fracture. Following orthopedic and biological evaluations, hip radiography and bacteriological examination, necrosis of the femoral head with a “cut-out” phenomenon was diagnosed as a possible complication of the previous glucocorticoid therapy (Figure 2d). Vancomycin and clindamycin-sensitive Staphylococcus epidermidis and Klebsiella were identified in the fistula secretion and treated with antibiotics. Due to this septic situation, the ablation of the implant and the surgical sanitation of the hip septic foci were mandatory. Meanwhile, the patient maintained spontaneous INR values of 3–5, which was difficult to manage in the perioperative septic setting. Lastly, the ablation of the osteosynthesis material and the Girdlestone-type femoral head resection arthroplasty were performed (Figure 2e). No intra- or periarticular purulent collections were detected intraoperatively. False membranes from inside the implant were collected and the same staphylococcus epidermidis was identified.

In October 2019, a follow-up abdominal multiphasic contrast-enhanced CT showed small-scale tumor progression and the patient was scheduled for surgery again (Figure 3e). The echocardiography performed during the preoperative cardiologic evaluation revealed hypokinesia of the ventricular septum; akinesia of the basal segment of the inferior wall; akinesia of the medium segment of the inferolateral, lateral and anterior walls; and severe left ventricular dysfunction with an ejection fraction of 28%. Coronary angiography uncovered trivascular atherosclerotic disease, and a percutaneous angioplasty with stent implantation was performed. The surgical intervention was postponed again.

Regrettably, in the spring of 2020 the COVID-19 pandemic reached our country and follow-up was possible only in July 2020, when, despite clinical control, the tumor (52/45 mm) and the peritumoral nodules had progressed; however, no hepatic or bone metastases were detected during morphological imaging (Figure 3f). As expected, we attempted surgery once more. This time, the patient was infected with SARS-CoV-2 and, by an unexpected fortunate chance, he had a completely asymptomatic infection.

In January 2021, a follow-up CT reconfirmed locoregional tumor progression without apparent distal metastasis (Figure 6a–c), and a distal pancreatectomy and splenectomy were performed with a favorable outcome. An open approach with a bilateral subcostal incision was chosen. After opening the abdomen, a thorough examination of the peritoneal cavity was performed. We discovered a 6/5 cm tumor in the tail of the pancreas, without any vascular or retroperitoneal involvement. No liver or peritoneal metastases were present. We opted for a distal spleno-pancreatectomy in a standard fashion with a clockwise approach and individual splenic artery and vein ligation, ensuring a proper oncological safety margin on the pancreatic body. The pancreas was transected using monopolar cautery (Figure 7a) with pancreatic duct identification and clipping (Figure 7b), and the pancreatic stump was double sutured: the first was an overlapping horizontal mattress suture, and the second was a figure-eight suture (Figure 7c). Two drainage tubes were placed, the abdomen was closed, and the specimen was sent for pathology examination (Figure 7d). The postoperative course was uneventful. We checked the amylase level in the drained fluid on the third postoperative day and it was normal, which allowed us to exclude the pancreatic fistula. The abdominal ultrasonography performed did not show any collections, and the drainage tubes were removed. The patient was discharged on the eighth postoperative day.

The histopathological aspects showed a well-differentiated, intermediate-grade (G2) pancreatic NET composed of cuboidal cells with mild-to-moderate cellular atypia, centrally located nuclei, eosinophilic granular cytoplasm, a trabecular growth pattern, a scant stromal reaction, calcifications, no necrosis and rare mitotic figures. The immunohistochemistry (IHC) was performed on 3 µm sections from 10% formalin-fixed paraffin-embedded tissues. According to the IHC method, an indirect bistadial technique was performed with a polymer-based detection system (UltraVision Quanto Detection System HRP DAB, Thermo Fischer Scientific, USA). The tissue sections were spread on poly-L-lysine-coated slides, immersed in three changes of xylene and rehydrated using a graded series of alcohol. Antigen retrieval was performed in a microwave oven. In each section, endogenous peroxidase was blocked by 10 min incubation in 3% hydrogen peroxide. The sections were incubated for 30 h with primary antibodies: CgA (Leica, clone 5H7, RTU), synaptophysin (Roche, clone SP11, RTU), glucagon (Cell Marque, clone259A-15, 0.5 mL, dilution 1:500), CD56 (Roche, clone 123C3mAb, RTU), Ki67 (Roche, clone 30-9, RTU), SSTR2 (Abcam, monoclonal, dilution 1:500), SSTR5 (Abcam, monoclonal, dilution 1:500). The UltraVision Quanto Detection System HRP was then applied for 10 min. Finally, the sections were incubated in 3’3-diaminobenzidine for 5 min, counterstained with Meyer´s hematoxylin and mounted. The slides were examined and photographed with a Leica DM750 microscope. Negative controls were obtained by replacing the primary antibody with non-immune serum. As a positive control a pancreatic tissue section was used. The IHC assay showed that the tumoral cells expressed glucagon, CgA, synaptophysin, CD56, SSTR2 and SSTR5 and the Ki67 proliferation index was 10% (Figure 8).

The 3-month postoperative CgA, pancreatic polypeptide and glucagon plasma levels (193 ng/L) were normal. A follow-up CT revealed a few millimetric calcifications in the right lobe of an otherwise homogenous liver and a 105/53/48 mm fluid-filled encapsulated collection in the spleno-pancreatic space with contrast enhanced thin walls and surrounding surgical clips. Up to 40% of patients have fluid collections after distal pancreatectomy as a result of a small pancreatic leak, yet they usually are without clinical significance and very few of them will require percutaneous drainage. Since the patient was symptom-free, we decided to wait for a spontaneous resolution of the collection. No local tumor recurrence was detected. There was a slight enlargement of the lymph nodes in the perihilar hepatic region, a 15/13 mm portocaval adenopathy and an 8/7 mm newly-appeared nodule on the fissure of the round hepatic ligament (Figure 6d–f). Our MDT decided to resume SSA therapy and follow-up with the patient at 6 months.

## 3. Discussions

Glucagonomas are rare functional pNETs for which Stacpoole developed the mandatory diagnostic criteria in 1981 that include the presence of a pancreatic tumor, positive immunochemical staining for glucagon in the majority of tumoral cells or increased tissue levels of immunoreactive glucagon, elevated basal plasma glucagon levels and at least one of the following manifestations: skin rash, glucose intolerance or/and hypoaminoacidemia [19]. Nowadays, increased plasma glucagon levels above 500 pg/mL accompanied by hyperglycemia are highly suggestive of a positive diagnosis. Moreover, even if plasma glucagon levels are below 500 pg/mL, in the presence of other evocative clinical symptoms, one should not exclude a glucagon secreting tumor. Thus, a biopsy for histological examination is no longer compulsory for a conclusive diagnosis [4].

Even with these simplified diagnostic criteria, the average time between symptom onset and the diagnosis of glucagonoma is 31.4 months [20]. The clinical features of glucagonoma syndrome are non-specific; consequently, many patients are misdiagnosed or diagnosed late. In their review, Song et al. highlighted the diversity of clinical findings and their incidence as follows: NME (82.4%); diabetes (68.5%); weight loss (60.2%); anemia (49.6%); and glossitis, stomatitis or cheilitis (41.2%). Almost half of the patients had metastatic disease upon diagnosis [20]. In a retrospective study conducted on 21 patients from Mayo Clinic, all with metastatic disease at presentation, Wermers et al. concluded that the leading sign was weight loss (71%), followed by NME (67%), diabetes mellitus (38%), cheilosis or stomatitis (29%) and diarrhea (29%). They also reported that while only 8 out of 21 patients had diabetes mellitus at presentation, 16 patients eventually developed diabetes. NME was never present before diabetes and, of utmost importance, the combination of these two disorders led to a more rapid diagnosis (7 months) than either symptom alone (4 years) [11].

Although not pathognomonic, NME is the sine qua non for glucagonoma syndrome. Most often, NME is the presenting sign; however it may occur any time during the course of the illness [21,22,23]. Given the rarity of the condition and the nonspecific clinical and pathological findings, misdiagnosis as other chronic dermatoses (mainly inverse psoriasis, eczema, blistering disorders, zinc, essential fatty acid or vitamin B complex deficiencies) is frequent and the diagnosis is delayed for months, or even years. Multiple biopsies, taken from the lesions’ active border, are usually necessary for diagnostic confirmation. Nevertheless, the histopathologic changes differ according to the stage of the lesions (from dyskeratotic dermatitis in early lesions to confluent parakeratosis and the necrosis of the upper spinous layers in later stages), may only be observed focally and cannot be distinguished from zinc or niacin deficiency and necrolytic acral erythema [24,25]. In our patient, the onset of the characteristic cutaneous lesions took place almost a decade after the diabetes diagnosis and months after the occurrence of the deep vein thrombosis, an association that could have pinpointed the correct diagnostic. Unfortunately, by the misinterpretation of the skin rash as pustular psoriasis, the diagnosis of glucagonoma was further delayed. The lack of response to several otherwise adequate treatment regimens for pustular psoriasis led to the reassessment of the patient and reconsideration of the dermatologic disease. Eventually, the typical refractory rash represented an important clue for establishing the correct diagnosis, simplified by its association with other signs and symptoms suggestive of glucagonoma.

The pathogenic mechanisms underlying the development of NME are incompletely elucidated. Hyperglucagonemia is unquestionably a crucial factor in NME pathogenesis, as the surgical removal of the glucagonoma or the initiation of treatment with glucagon antagonists promptly leads to the remission of NME [26,27]. This was also the case in our patient, who experienced rapid amelioration of NME after treatment with long-acting somatostatin analogues was started. Nevertheless, the typical skin manifestations are not encountered in other disorders associated with high glucagon levels, such as acute trauma, hepatic cirrhosis, pancreatitis, renal insufficiency, celiac disease, inflammatory bowel disease, Cushing’s syndrome or familial hyperglucagonemia [28]. Moreover, glucagonoma patients may experience the amelioration or even resolution of the skin eruption despite a constantly increasing glucagon level [29]. Therefore, hyperglucagonemia is necessary, but not sufficient, for the epidermal dysfunction characteristic to NME. Multiple interconnected factors are suspected to play important pathogenic roles, among which amino acid, zinc and essential fatty acid deficiencies are essential [29]. The correction of these nutritional deficiencies results in improvement of NME [30,31]. In addition, hepatic dysfunction promotes the development of NME, as the liver is responsible for glucagon degradation [32]. The high plasma glucagon level generates a catabolic state that explains the associated hypoproteinemia and hypoaminoacidemia. It also stimulates hepatic gluconeogenesis by means of amino acid catabolism. The amino acid shortage causes diminished tissular peptide synthesis with subsequent increased epidermal turn-over and dysfunction [28]. In addition, it impairs the binding of zinc and essential fatty acids to serum albumin and, implicitly, their transport and tissular distribution. Even minor variations of extracellular zinc levels significantly influence the function of numerous cell types, including keratinocytes, as it modulates protein, lipid and nucleic acid metabolism. Zinc deficiency favors essential fatty acid deficiency, given that it is indispensable for the desaturation of linoleic acid. Furthermore, glucagon stimulates peripheral lipolysis, worsening essential fatty acid deficiency. Since the production of essential fatty acids is hindered, their precursors are deviated to the arachidonic acid pathway, consequently facilitating the synthesis of proinflammatory prostaglandins and leukotrienes [30]. Taking this complex interplay into consideration, the multifactorial malnutrition model is currently the generally accepted concept regarding the pathogenesis of glucagonoma syndrome.

Up to 50% of glucagonoma cases are associated with a venous thrombosis that usually manifests as deep vein thrombosis or pulmonary embolism [4]. The pathophysiology of the glucagonoma prothrombotic state is attributable to a molecule similar to coagulation factor X, which is produced by the tumor cells [33]. Immobility is a well-known major risk factor for thrombosis [34]. Our patient had an extensive left leg deep vein thrombosis, he was unable to walk unassisted and had an erratic response to vitamin K antagonists associated with exceedingly unpredictable INR levels on the same dosage. Moreover, he briskly developed sagittal sinus thrombosis after intravenous phytonadione administration in the attempt to lower the high risk of bleeding assigned by a high INR value and, afterwards, maintained spontaneous elevated INR levels in septic conditions.

Owing to its positive inotropic and chronotropic properties, glucagon was historically used as treatment for severe beta-blocker overdose in two cases reported by Peterson et al. Circulating supra-physiological levels of glucagon may be responsible for resting tachycardia or other cardiac impairments seen in glucagonoma syndrome [35]. Indeed, on rare occasions, glucagonoma may be associated with cardiomyopathy, acute pulmonary edema or left ventricular failure [20]. In one case, reported by Chang-Chretien et al., a 54-year-old woman presented with congestive heart failure, sinus tachycardia, dilated cardiomyopathy with an ejection fraction of 15%, and a coronary angiography ruled out a secondary cause for the cardiac disease [36]. In a more recent glucagonoma case report, Barabas et al. documented a series of 4-year follow-ups with echocardiography of a progressive left ventricular dilatation and dysfunction, in the absence of ischemia, that did not respond to SSA [37]. However, our patient, in the presence of multiple cardiac risk factors and elevated glucagon levels, developed severe ischemic coronary disease at a young age.

The majority of pNETs are well-differentiated, low-grade tumors that have a mitotic count less than 2/10 high-power field (HPF) and/or a Ki-67 index of less than 3% [18]. Glucagonoma has a good prognosis, with a relative 5-year survival rate of 75% if diagnosed in stage 1. On the other hand, having a slow growth rate and non-specific clinical features, it is rarely diagnosed in an early stage and over 50% of patients are metastatic at the time of diagnosis [4,5]. The most common site of metastasis is the liver, followed by the regional lymph nodes, bones, adrenal glands, kidneys, and lungs [4]. In our case, the tumor was well-differentiated at an intermediate grade, with a presumable evolution of 10 years until he was referred to our endocrinology department (if we assign the diagnosis of diabetes mellitus as the onset of the disease). Interestingly enough, our patient did not present hepatic metastases at the time of diagnosis, and he did not develop distant metastases for 3 years. As frequently encountered, the pathological and immunochemistry analyses from EUS-FNA yielded slightly different results from those performed after the surgical removal of the tumor. Initially, the tumor was described as low-grade and well-differentiated with a Ki 67 index of 2%, and afterwards as well-differentiated and of an intermediate grade with a Ki 67 index of 10%. In one study of 106 tissue biopsies obtained using EUS-FNA, Sudhoff et al. evaluated the diagnostic accuracy of this method. EUS-FNA showed an overall sensitivity of 78% and a specificity of 100%, while the accuracy was greater for mediastinal and retroperitoneal lesions, up to 89%. The accuracy for pancreatic lesions and perigastric lymph nodes was significantly smaller (<80%) [38]. Analyzing data from 91 patients with pancreatic tumors, Sheng-ShunYang et al. showed similar results for EUS-FNA diagnostic accuracy (79.2%) [39]. As in our case, EUS-FNA remains a valuable diagnostic tool, especially when surgery is not feasible; however, it may yield discrepant results, mostly in settings with a limited amount of diagnostic material for an accurate tumor grading.

There are many useful imaging methods used in pNET evaluation, staging and follow-up. Although 18FDG PET/CT is better suited for G3 and high grade G2 pNETs, it can also be helpful in G1 and low-grade G2 evaluation, especially in combination with SRS [40]. In our case, a particular aspect of tumor appearance on both molecular imaging modalities, 18FDG-PET/CT and SRS SPECT/CT, consisted of the high uptake of both radiolabeled SSA as well as the glucose analogue. A possible explanation for this feature could reside in the heterogeneity of the tumor phenotype, which is difficult to assess in the pathologic specimen. The immunohistochemical assessment of the percentage of cells proliferating and the Ki67 index is determined in hotspot areas of the resected tumors or biopsies. However, intra-tumoral and, in the case of disseminated disease, intra-patient heterogeneity in the tumor phenotype can easily introduce an erroneous interpretation of disease aggressiveness [41,42]. Low and intermediate grade tumors are particularly challenging when it comes to risk stratification. Johnbeck et al. investigated 88 patients with well-differentiated advanced NET; the 18FDG PET was positive in 39% of G1 and 50% of G2 patients. Moreover, FDG-positivity was a significant prognostic factor with a hazard ratio of 2.4 for progression free survival (PFS) and 5.3 for overall survival (OS) [43]. In another study, Binderup et al. evaluated 98 patients with advanced NET. The 18FDG PET was positive in 40% of G1, 70% of G2 and 93% of G3 patients. A total of 14 patients died, of whom 13 had a positive FDG scan (hazard ratio of 10.3). Among the 47 patients with G1 NET, five died and four of them were FDG-positive [44]. Koch et al. evaluated the use of Ga-68-DOTA-TATE and SUVs to predict the effectiveness of treatment with octreotide acetate in 30 patients with well-differentiated NETs of the ileum. For each of them, they determined the maximum SUV (SUV max) and the average SUV of a 50% isocontour volume of interest covering the lesion with maximum uptake (SUV mean); they found that SUV max and SUV mean were important prognostic indices for predicting the response to therapy with octreotide acetate. The cut-off values of SUV max and SUV mean that separated patients with a long PFS (69.0 weeks; 95% CI 9.8–128.2) from those with a short PFS (26.0 weeks; 95% CI 8.7–43.3) response to SSA therapy were 29.4 and 20.3, respectively [45]. In a retrospective study Ambrosini et al. collected data from 43 patients with G1 or G2 pNET. They noted that SUV max was significantly higher in patients with stable disease versus those with progressive disease, and concluded that this represents a relevant prognostic factor in patients with G1 and G2 pNET that should be routinely used for disease characterization and patient management [46]. The importance of the dual-tracer evaluation was highlighted by Chan et al. when they proposed the NETPET scoring system. After both imaging studies are performed, the tumors are categorized as it follows: P1 indicates purely somatostatin receptor-positive lesions without FDG uptake above background. P2–4 lesions are positive for both scans but with a progressive increase in FDG uptake (relative to uptake on SRS) from P2 to P4. P5 indicates lesions that are positive for FDG and negative for SRS. Although OS was significantly associated with NETPET grade, categories 2–4 were merged due to the limited number of patients investigated (62). Thus, we still need more data for an accurate interpretation of this intermediate category in relation to prognosis [47]. The NETPET scoring system may prove useful in therapeutic decisions. For example, high uptake of 68Ga-DOTATATE has been associated with a better response from SSA, while patients with positive FDG PET/CT and negative SRS are good candidates for chemotherapy. The benefit of PRRT alone or coupled with chemotherapy could additionally be assessed by such a scoring system [40]. A prospective study published by Binderup et al. including 166 GEP-NEN patients (140 with G1 and G2 tumors) concluded that positivity on ^18^F-FDG-PET imaging is a powerful prognostic tool of relevance for patients with all grades of NEN [48].

Another important topic is the screening for genetic alterations known to increase the risk for pNET occurrence [4]. Mutations in DNA repair genes, such as MUTYH, CHECK2 and BRCA2, together with MEN1 and VHL mutations were found in 17% of patients with clinically sporadic pNET [6]. About 70% of the patients diagnosed with MEN1 syndrome were found to harbor mutations in the MEN1 gene; however, alterations in CDKI genes were also demonstrated [49]. Our patient had no family or personal history suggestive of a tumoral syndrome and the biological screening for MEN1 syndrome was negative.

For the treatment of a TNM stage III, well-differentiated, intermediate grade G2, sporadic, functional pNET, located in the pancreatic tail with regional lymph node involvement, the guidelines recommend distal pancreatectomy plus peripancreatic lymph node dissection and en bloc splenectomy, which we finally accomplished. Meanwhile, SSA therapy with lanreotide was not only efficient in suppressing tumor aberrant hormonal secretion, but likely also in restraining tumoral growth [15,18]. SSAs are an established antiproliferative therapy for intestinal NETs and can also be recommended for the prevention or inhibition of tumor growth in both intestinal and pancreatic NETs. In a PROMID trial, Rinke et al. showed that SSA therapy with octreotide improved the time to tumor progression (TTP) (14.3 vs. 6 months) in carcinoid tumors of the midgut [50]. Moreover, the current guidelines recommend lanreotide Autogel as a first-line systemic therapy for pNETs with a Ki-67 index < 10% [51]. The data from CLARINET and CLARINET OLE studies, in which 120 mg lanreotide Autogel every 28 days demonstrated a 53% risk reduction in disease progression or death and a significantly prolonged median PFS over deferred treatment; the median PFS benefits were observed irrespective of tumor origin, tumor grade and hepatic tumor load. Furthermore, tumor control was achieved as early as 12 months without altering QoL and sustained PFS benefits were maintained with prolonged treatment. The CLARINET OLE study showed a favorable long-term safety and tolerability profile of lanreotide Autogel during a median treatment duration of 40 months. Furthermore, switching from a placebo to lanreotide was associated with an estimated 14.0-month time to progression and similar adverse effects to those in the OLE and core CLARINET studies [52,53]. In a recently published paper, Merola et al. evaluated 73 patients (61 with liver metastases; 68 G2, 5 G3) with well-differentiated pNETs and a Ki-67 index ≥ 10% receiving long-acting SSAs between 2014–2018. As expected, the increased tumor grade and hepatic tumor load were independently associated with shortened PFS. They did not note any statistically significant difference between octreotide and lanretotide on PFS. Lastly, they demonstrated that SSA exerted antiproliferative activity against intermediate and high grade pNETs, with both octreotide LAR and Lanreotide being associated with a median time to next treatment of 14.2 month and a median PFS of 11.9 months [54]. Although mild adverse effects, such as hyperglycemia and pancreatic enzyme insufficiency, could have been associated with SSA therapy, our patient responded very well with resolution of diarrhea, complete remission of the NME and significant metabolic improvement.

## 4. Conclusions

As in other functional pancreatic NETs, early clinical recognition of hormonal hypersecretion syndrome and the multidisciplinary approach are the keys for best patient management.

## Figures and Tables

**Figure 1 diagnostics-12-00216-f001:**
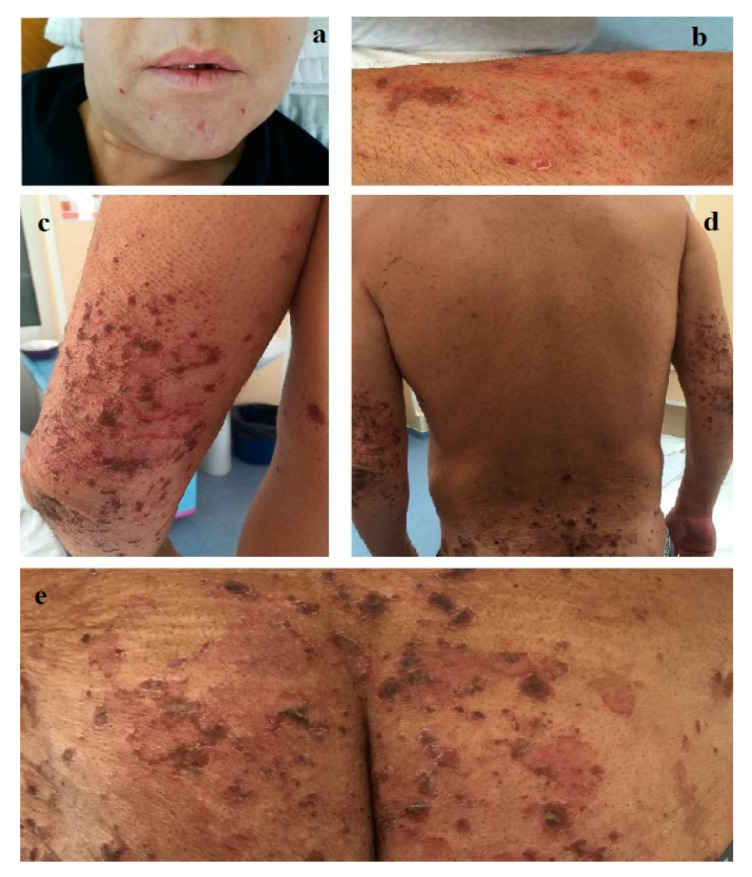
(**a**) Cheilitis and perioral eroded papules (scale 1:8). (**b**) Red papules, papulo-vesicles and eroded plaques affecting the lower limb (scale 1:4). (**c**–**e**) Irregular erosive plaques, showing slightly elevated pustular and scaling borders, some covered with crusts, with a tendency to coalesce into large, circular eroded areas distributed on the upper limbs (scale 1:5), lumbosacral area (scale 1:10) and buttocks (scale 1:4).

**Figure 2 diagnostics-12-00216-f002:**
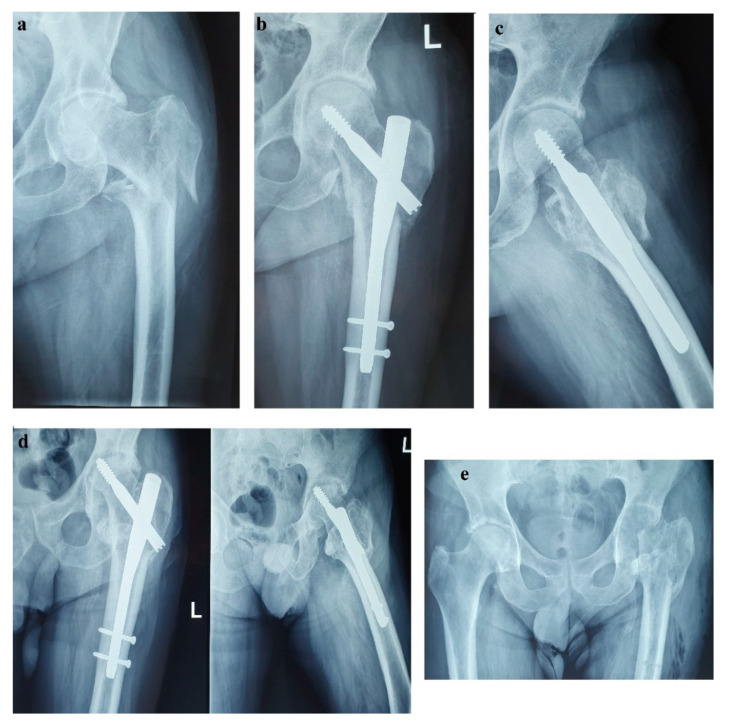
(**a**) Left (L) pertrochanteric fragility fracture; (**b**,**c**) Gamma nail osteosynthesis; (**d**) Femoral head osteonecrosis and cut-out complication; (**e**) Girdlestone resection arthroplasty.

**Figure 3 diagnostics-12-00216-f003:**
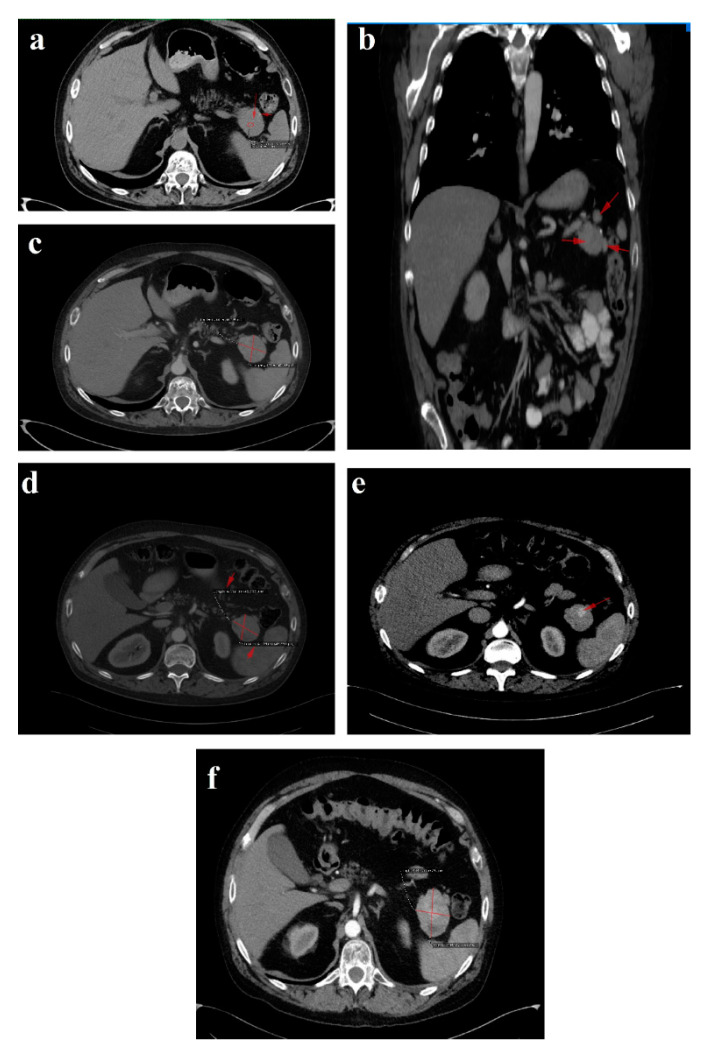
1 February 2018 CT scan: (**a**) axial image without contrast showing a round solid mass with microcalcification in the tail of the pancreas; (**b**) arterial phase coronal reconstruction showing a hypervascular mass with satellite nodules situated in the pancreatic tail; and (**c**) axial image, arterial phase showing a 35/44 mm tumor in the pancreatic tail. September 2018 follow-up CT: (**d**) axial image, arterial phase showing stable disease. October 2019 follow-up CT: (**e**) axial image, arterial phase showing inhomogeneous contrast enhancement and small-scale tumor progression. July 2020 follow-up CT: (**f**) axial image, arterial phase showing evident tumor mass progression.

**Figure 4 diagnostics-12-00216-f004:**
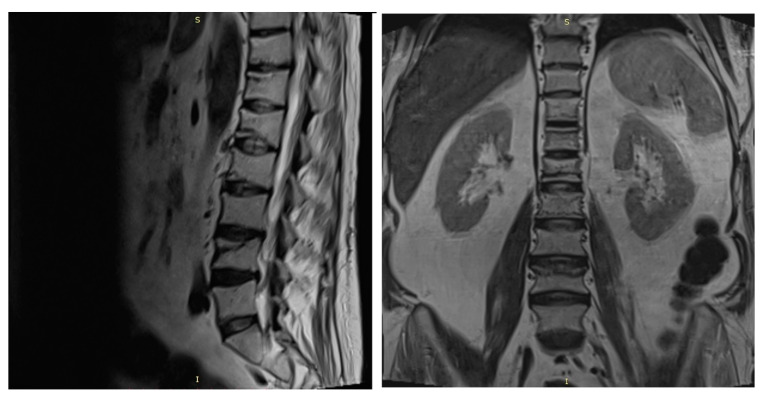
Spine MRI showing T11, T12, L1, L2 and L4 vertebral fractures. (s = superior, I = inferior).

**Figure 5 diagnostics-12-00216-f005:**
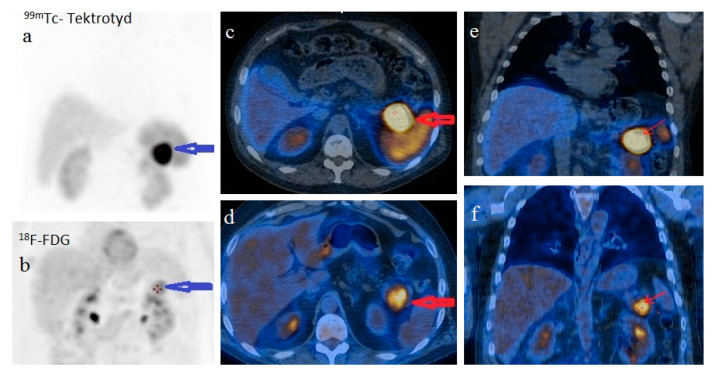
Head-to-head comparison of ^99m^Tc-EDDA/HYNIC-TOC SPECT/CT SSTR scintigraphy and ^18^F-FDG PET/CT on the (**a**,**b**) MIP images and (**c**,**d**) fused axial images with high uptake of both radiotracers (radiolabeled somatostatin analogue and glucose analogue, respectively) in the pancreatic tail tumor (arrows) and no uptake in the satellite tumor (described on contrast-enhanced CT) or elsewhere on whole body images. ^99m^Tc-EDDA/HYNIC-TOC SPECT/CT SSTR scintigraphy and ^18^F-FDG PET/CT fused coronal images: (**e**,**f**) higher uptake of radiolabeled somatostatin analogue compared to glucose analogue in the pancreatic tail tumor (arrows).

**Figure 6 diagnostics-12-00216-f006:**
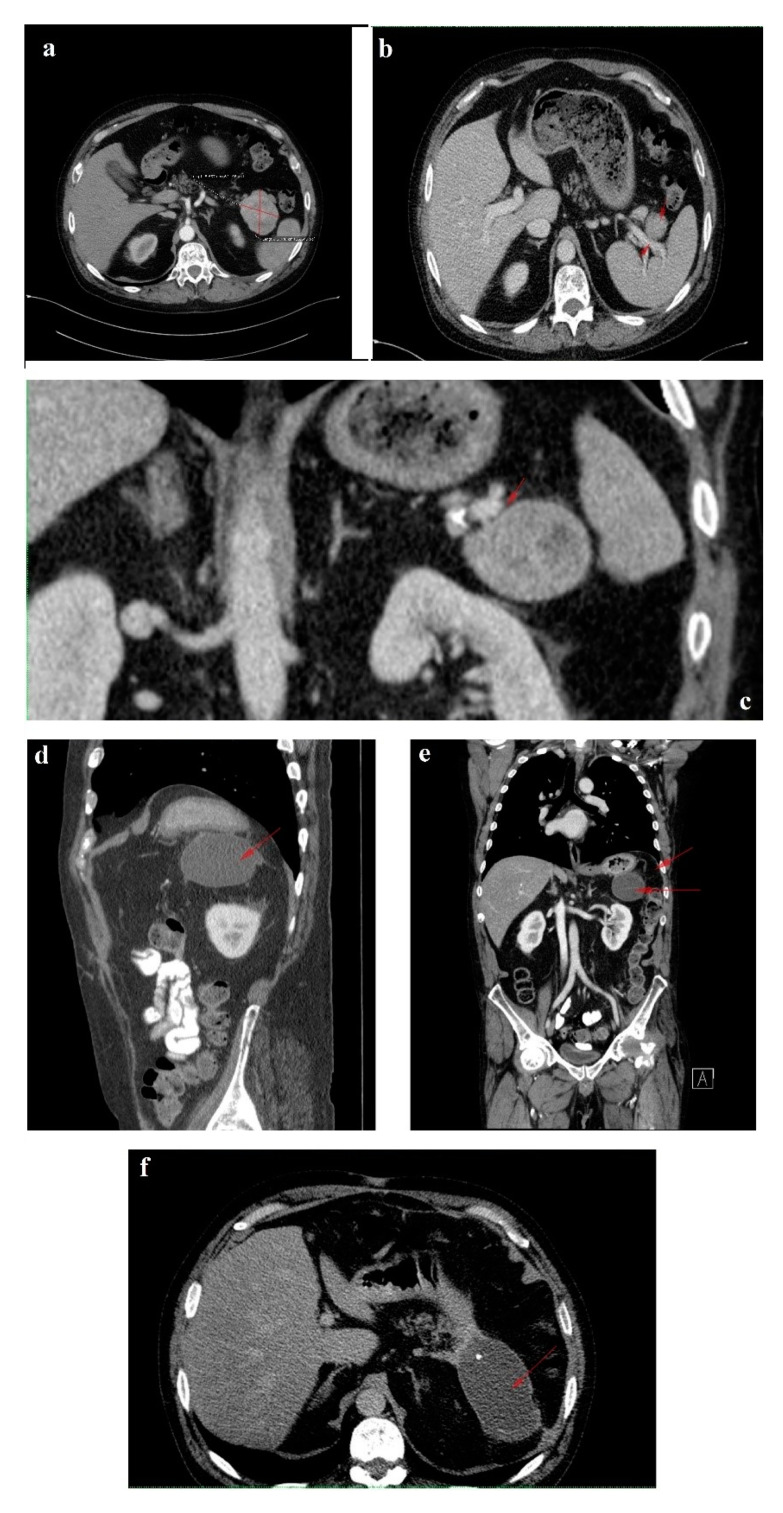
Follow-up CT January 2021: (**a**) axial image-tumor progression; (**b**) axial venous phase with no vascular invasion; (**c**) coronal reconstruction venous phase, shows the fat plan between the mass and the vascular structures, indicating that the lesion is operable. Follow-up CT April 2021: (**d**–**f**) venous phase sagittal, coronal and axial reconstructions, respectively, showing fluid collection after the surgical removal of the pancreatic masses and the spleen. Red arrows and line indicate the lesions described in the figure legend; further information may be redundant.

**Figure 7 diagnostics-12-00216-f007:**
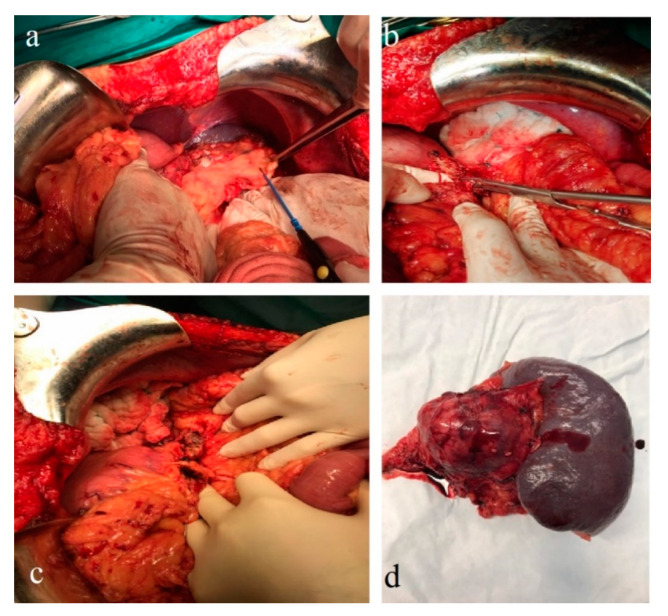
Intraoperative images: (**a**) pancreatic transection; (**b**) pancreatic duct isolation and clipping; (**c**) pancreatic stump closure; and (**d**) the pathology specimen.

**Figure 8 diagnostics-12-00216-f008:**
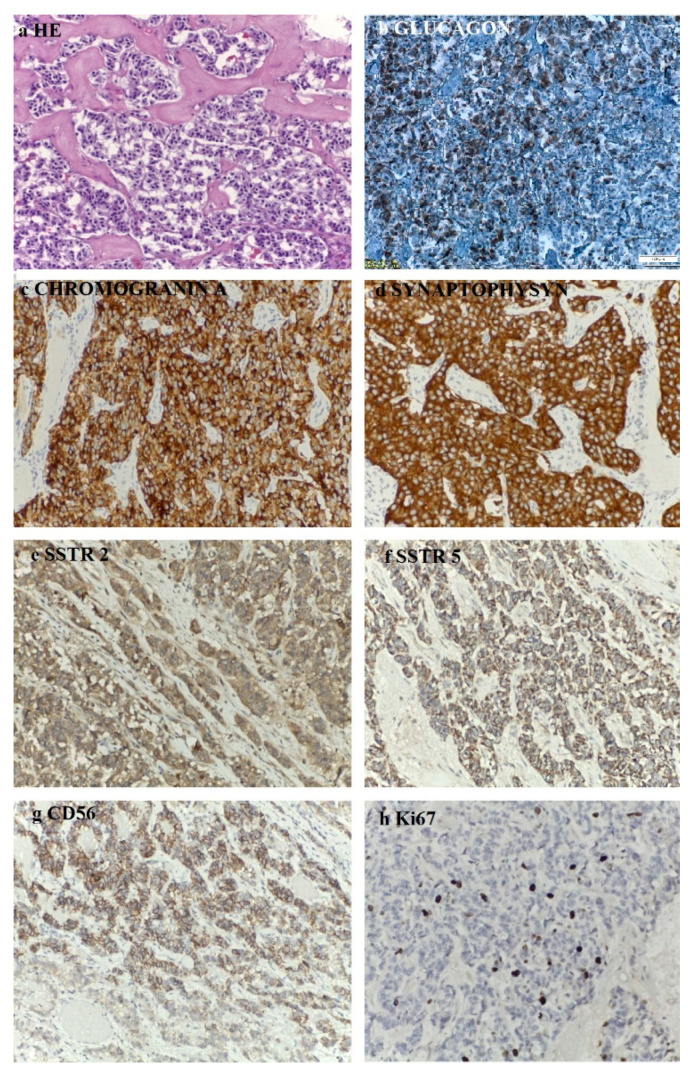
Histopathological (20×) and immunohistochemistry (20×) findings: (**a**) well-differentiated Pan-NET, G2, with mild to moderate cellular atypia and trabecular architecture; (**b**) glucagon (Cell Marque, clone259A-15, 0.5 mL, dilution 1:500)-positive reaction in the tumoral cells, cytoplasmic, with internal positive control; (**c**) CgA (Leica, clone 5H7, RTU)-positive reaction in the tumoral cells, cytoplasmic, with internal positive control; (**d**) synaptophysin (Roche, clone SP11, RTU)-positive reaction in the tumoral cells, cytoplasmic, with internal positive control; (**e**) SSTR2 (Abcam, monoclonal, dilution 1:500)-positive reaction in the tumoral cells, cytoplasmic, with internal positive control; (**f**) SSTR5 (Abcam, monoclonal, dilution 1:500)-positive reaction in the tumoral cells, cytoplasmic, with internal positive control; (**g**) CD56 (Roche, clone 123C3mAb, RTU)-positive reaction in the tumoral cells, cytoplasmic, with internal positive control; (**h**) Ki67 (Roche, clone 30-9, RTU) proliferation index was 10% in tumoral cells, with internal positive control.

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
