# Peer review of "How Many Times Can One Go Back to the Drawing Board before the Accurate Diagnosis and Surgical Treatment of Glucagonoma?"

_diagnostics, 2022, doi:10.3390/diagnostics12010216_

Round 1

Reviewer 1 Report

This manuscript discusses a case report of a patient with glucagonoma and gives a narrative review on the issue.

The article is interesting to read, sufficiently structured and well written but needs further minor editing in writing and grammar.

Some comments:

Title: why is there a hyphen after the question mark?

Authors: An “and” seems to be too much after the last author

Abstract:

Line 36: abnormal glucagon secretion instead of glucagon abnormal secretion

Line 45: Enlarge CT when it is firstly mentioned

Intro:

Line 65: At diagnosis instead of presentation

Line 97: The mechanistic reasons for weight loss and thrombembolism could be shortly explained

Line 126: In patients with limited liver tumor burden hepatic resection is recommended. Please explain this further

Case presentation:

Line 143: A widespread instead of “an”

Line 151: Which diagnostics led to the diagnosis of psoriasis?

Line 174: Gamma nail instead of Gama nail

Line 185: Was CRP available?

Line 209: The sentence seems incomplete

Line 214: Was Calcium and Vitamin D also introduced?

Line 256: Klebsiella instead of Klebsiela

Line 339: CgA is partly written different. Please unify

 In general, I would suggest this manuscript to be submitted as review (not as case report) in order not to mislead the reviewing person. 

Author Response

Thank you very much for the interest in our paper and for the valuable suggestions.

As you suggested, we decided to slightly change the title and to submit the manuscript as narrative review.

The “ and” after the last author was removed.                                            

Line 36: we changed glucagon abnormal secretion with abnormal glucagon secretion

Line 45: we enlarged CT when it is firstly mentioned

Line 65: we changed At diagnosis instead of At presentation

Line 97, 126- we shortly explained the mechanistic reasons for weight loss and thromboembolism

and the reasons for hepatic resection; they are also discussed in the discussion section

Line 143: we changed A widespread instead of “an”

Line 151: the same skin lesions led to the erroneous diagnosis of psoriasis

Line 174: we corrected Gamma nail instead of Gama nail

Line 185: we added CRP level at diagnosis

Line 209: we reformulated the sentence

Line 214: we added calcium, 25 OH vitamin D, PTH, PRL and IGF1 values  performed as biochemical screening for MEN1, which was negative

 Line 256: we corrected Klebsiella instead of Klebsiela

Line 339: we corrected all CgA

Reviewer 2 Report

COMMENTS

Title: How many times can one go back to the drawing board before the surgical treatment of glucagonoma?- case presentation and review of the literature

  1. While the title is descriptive of the difficulties that were encountered in this patient, it fails to completely capture the uniqueness of the case .

Maybe a better title would be, “Difficulties in the diagnosis and management of glucagonoma: A Case report.” The current title is also possible but include “Case Report” in the title e.g. “How many times can one go back to the drawing board before the surgical treatment of glucagonoma: A Case Report?

  1. General comments
  2. The entire article is well appreciated as it presents to the reader difficulties in the diagnosis (delayed in this case because of a previous misdiagnosis) and management of a case of glucagonoma especially in the context of the pandemic. However, please do a grammar and spelling check as there are a few words which are not used in its proper meaning e.g. on page 7 second paragraph “Imagistic”; the correct word is “imaging” proof of systemic metastases. Also on the first sentence of page 8, “peripheric” – the correct word is “peripheral”.
  3. The paper is also quite long for a case report e.g. total of 19 pages, 16-17 pages of the main write-up. For example, the “introduction” consists of 2 pages of 8 paragraphs when typically the Introduction according to the CARE guidelines may just be 2-3 paragraphs that answers the questions: “What is unique about this case and what does it add to the scientific literature?” The rest of the information may be placed in the discussion.

For example, the authors could highlight the difficulties in the diagnosis in this case because the typical skin lesions were mistaken to be from other diseases. Delays in the management because of the complications from the skin lesions (infection) and other complications arising from the treatment (glucocorticoid-induced osteoporosis causing fragility fracture), and delays in surgery due to COVID itself (e.g. patient became infected with COVID plus  surgical delays due to scheduling).

Another way to shorten the paper is through more concise writing. For example, at the bottom of page 4 (last paragraph) is written: “Unfortunately, in October 2017, the patient fell from his standing height, suffered a left pertrochanteric comminuted fragility fracture and osteosynthesis with Gama nail was performed (Figure 2 a- c). The immediate evolution was further complicated by wound dehiscence that called for surgical debridement.” It can be shortened, “In October 2017 the patient suffered a left hip trochanteric comminuted fragility fracture after a fall from standing height. Gama nail osteosynthesis surgery was performed (Figure 2 a-c) but unfortunately, this was complicated by wound dehiscence requiring surgical debridement.” From 43 words to 39 words. If this can be done similarly for the rest of the manuscript, then it will shorten it significantly.

  1. Figure 2 may be removed since the fracture is a complication of the prolonged glucocorticoid use and including the pictures of the osteosynthesis and Girdlestone resection do not really add much to the case report of glucagonoma.
  2. In the conclusions, please re-emphasize the lessons learned especially that in the presence of the pathognomic NME skin lesions coupled with systemic manifestations of diabetes, anemia and weight loss, and oral lesions (cheilitis, stomatitis or glossitis), that a diagnosis of glucagonoma should be considered. Timely diagnosis can lead to definitive therapy in the form of surgical resection.

Author Response

Thank you very much for the interest in our paper and for the valuable suggestions.

As suggested by the reviewers we decided to slightly change the title and, with your permission, to submit the manuscript as narrative review.

In the light of these considerations and with the hope to unite all the reviewer’s recommendations the manuscript was reconstructed as a narrative review and we added the information requested by the reviewers.

We did an extensive English language grammar and spelling check.

We corrected, as suggested, imagistic with “imaging” and “peripheric” with “peripheral”; also we corrected line 247 -characterization

As we do have an extensive discussion section we hope for  a clearly stated conclusion without redundant information.

This manuscript is a resubmission of an earlier submission. The following is a list of the peer review reports and author responses from that submission.